# BAYESIAN ENSEMBLE FOR SEQUENTIAL DECISION-MAKING

**Rui Liu**[1*], **Enmin Zhao**[1*], **Lu Wang**[1†], **Yu Li** [2†], **Ming Pang**[1], **Changping Peng**[1],
**Zhangang Lin**[1], **Ching Law**[1], **Jingping Shao**[1]

[1]JD.com, China
[2]Institute of Computing Technology, Chinese Academy of Sciences, China
{liurui34,zhaoenmin,wanglu241}@jd.com, liyu@ict.ac.cn,
{pangming8,pengchangping,linzhangang,shaojingping}@jd.com

## ABSTRACT

Ensemble learning is a practical family of methods for uncertainty modeling, particularly useful for sequential decision-making problems like recommendation systems and reinforcement learning tasks. The posterior on likelihood parameters is approximated by sampling an ensemble member from a predetermined index distribution, with the ensemble's diversity reflecting the degree of uncertainty. In this paper, we propose Bayesian Ensemble (BE), a lightweight yet principled Bayesian layer atop existing ensembles. BE treats the selection of an ensemble member as a bandit problem in itself, dynamically updating a sampling distribution over members via Bayesian inference on observed rewards. This contrasts with prior works that rely on fixed, uniform sampling. We extend this framework to both bandit learning and reinforcement learning, introducing Bayesian Ensemble Bandit and Bayesian Ensemble Deep Q-Network for diverse decision-making problems. Extensive experiments on both synthetic and real-world environments demonstrate the effectiveness and efficiency of BE.

## 1 INTRODUCTION

Sequential decision-making under uncertainty lies at the core of recommendation systems and reinforcement learning (RL) (Sutton & Barto, 1998), where agents must continually balance exploiting known high-reward actions against exploring uncertain options to maximize long-term returns. This exploration–exploitation trade-off is most famously distilled in the multi-armed bandit (MAB) problem. MAB is a foundational RL setting in which an agent sequentially pulls one of $K$ arms, and each arm is associated with an unknown stochastic reward distribution (Slivkins, 2019; Lattimore & Szepesvári, 2020; Bouneffouf & Rish, 2019; Bubeck & Cesa-Bianchi, 2012). In order to implement efficient exploration (select informative actions or visit different states), the agent must understand the "uncertainty" degree of its acquired knowledge. Therefore, uncertainty modeling often plays a central role for these tasks.

Thompson sampling (TS) follows this uncertainty modeling paradigm. It maintains a posterior distribution of the parameters of the likelihood function for all possible arms' rewards (Thompson, 1933; Li et al., 2010; Chapelle & Li, 2011; Kawale et al., 2015; Russo et al., 2018). This posterior distribution characterizes uncertainty of knowledge on arms. At each iteration, one sample of the likelihood parameters are drawn from the posterior distribution, and then the sample is utilized to select the "best" arm by calculating the reward expectation for each arm. After the actual reward is observed, the agent updates the posterior distribution.

A possible way to account for the complex relation of each arm's context and reward is to use neural networks for modeling the likelihood function. The key challenge lies in the fact that maintaining the exact posterior distribution is often computationally intractable under general circumstances, especially in the neural network case (primarily owing to entailing exact Bayesian inference). A series

---

[*]Equal contribution.
[†]Corresponding authors.

of works address this challenge by approximate Bayesian inference based on variational inference, Markov chain Monte Carlo, gradient-based methods and ensemble-based methods (Lu & Roy, 2017; Riquelme et al., 2018; Guo et al., 2020; Kveton et al., 2019; Zhang et al., 2021).

Among the methods discussed, ensemble learning is favored in practical applications due to its straightforward approach, versatility, and demonstrated effectiveness (Zhou, 2012). In ensemble learning, each iteration involves selecting a base model (an ensemble member) at random from a predetermined *index distribution* to predict the reward for each option. These base models represent samples from the prior/posterior distribution of the likelihood parameters, with their variety reflecting the uncertainty in predictions. Updating the base models' parameters with the received rewards is tantamount to conducting an approximate Bayesian inference on these distributions. Ensemble-based techniques have been shown to offer substantial benefits for joint predictions and enhance performance in sequential decision-making tasks when compared to other uncertainty-modeling agents. uncertainty (Snoek et al., 2019; Osband et al., 2022).

Current ensemble-based Thompson Sampling methodologies primarily concentrate on refining the model parameters of ensemble members to better approximate the posterior distribution. However, these methods maintain the sampling index distribution (such as a discrete uniform distribution) unchanged throughout the decision-making process. This approach does not fully account for the uncertainty inherent in the diversity of the ensemble members. In our setting, the performance of individual ensemble members similarly varies due to factors such as random initialization and the incorporation of prior functions. *Therefore, we suggest an update to both the parameters of ensemble members and the index distribution governing them.* The parameters are responsible for guiding the actions, while the index distribution is crucial in selecting the appropriate ensemble member. Essentially, we forge a direct nexus between the index distribution and the feedback rewards to achieve a more precise approximation of the posterior distribution.

Our contribution in this paper is two-pronged. Firstly, we propose Bayesian Ensemble (BE) — a unified framework adapting Bayesian index distribution updating to both contextual bandits and reinforcement learning, achieving state-of-the-art exploration efficiency through theoretically grounded variance reduction. This technique not only updates the parameters of ensemble members but also forges a direct link between the index and reward distributions by employing Bayesian inference to update the index distribution. BE's versatility allows it to enhance a broad spectrum of ensemble-based Thompson sampling methods and reinforcement learning methods. Secondly, we demonstrate the efficacy of BE in a variety of settings, including both synthetic and real-world environments, with a particular focus on recommendation systems within sequential decision-making problems. We conduct a thorough and meticulous comparison with cutting-edge deep Thompson sampling and ensemble reinforcement learning methods, providing a comprehensive benchmark that will aid in advancing future research in this area.

## 2 RELATED WORK

There is a long history in the research of uncertainty estimation aiming for the decision-making problem, of which the key challenge is balancing exploration and exploitation. Much of the work, as the extension of vanilla Thompson sampling (Thompson, 1933; Russo et al., 2018), maintains an approximate posterior distribution (used for posterior sampling) rather than making exact Bayesian inference. These methods include variational inference (Blundell et al., 2015), bootstrapping (Osband et al., 2016), MCMC (Welling & Teh, 2011; Ahn et al., 2012; Li et al., 2016), dropout (Gal & Ghahramani, 2016), ensemble sampling (Lu & Roy, 2017) and so on.

Among these promising posterior sampling techniques, ensemble sampling (ensemble-based TS) is preferable in practice owing to its simplicity and generality (Snoek et al., 2019). Variants of ensemble sampling based on neural networks has attracted considerable attention. To approximate posterior distribution, deep ensembles (Lakshminarayanan et al., 2017) yield predictive uncertainty estimates through multiple neural networks with random initialization. These ensemble members could further improve computing and storage efficiency through shared weights (Wen et al., 2020). Moreover, these ensemble-based approaches can be formalized as sampling an index from a fixed reference distribution and then mapping the index to an ensemble member. Deep ensembles (Lakshminarayanan et al., 2017) and ensembles with prior functions (Osband et al., 2018) sample neural networks from a uniform distribution. The index of hypermodel (Dwaracherla et al., 2020) is drawn

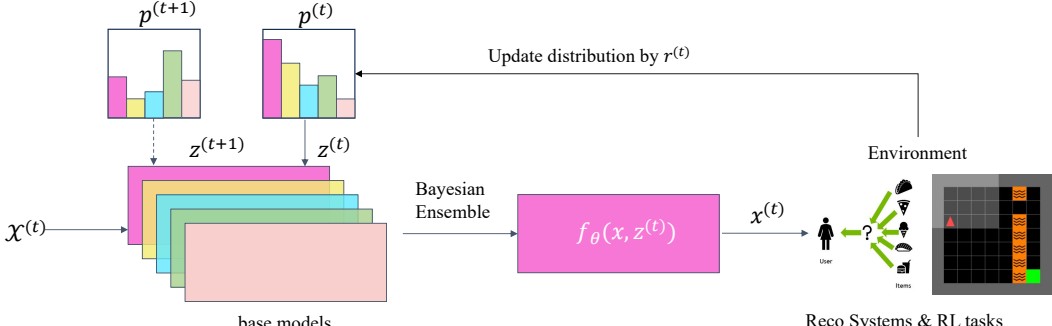

Figure 1: The Bayesian Ensemble (BE) framework. The agent maintains a probability distribution $p^{(t)}$ for the index $z \in \mathcal{Z}$ and bridges the gap between the index and reward distribution.

from a standard Gaussian distribution, and then is mapped to base model parameters with a neural network. Epistemic Neural Networks (Osband et al., 2023) maps a distributional index to an epinet which specially quantify uncertainty for the base network. Besides, Bayesian regret bounds of ensemble sampling for the linear bandit problem are theoretically analyzed (Qin et al., 2022).

Ensemble++ (Li et al., 2025) proposes a novel shared-factor ensemble architecture with random linear combinations. Different from Ensemble++, our method focuses on the dynamic Bayesian updating of index distributions. It models member selection as an inner-bandit problem, enhancing uncertainty modeling accuracy by directly linking indices to reward distributions.

Besides Thompson sampling, ensemble methods also play an important role in reinforcement learning. After the proposal of Deep Q-Network (DQN) (Mnih et al., 2015), Ensemble DQN (Anschel et al., 2017) is introduced to reduce the variance of the standard DQN. Ensemble DQN uses the average weighted combination of multiple value functions to guide policy updates, making the reinforcement learning process more stable. Inspired by Dropout, Random Ensemble DQN (Agarwal et al., 2020) proposes random weighting of multiple value functions to address the distribution mismatch problem in offline reinforcement learning. Reset Deep Ensemble (Kim et al., 2023), through the periodic reset operation on multiple ensemble agents, demonstrates its effectiveness in sample efficiency and safety considerations. HyperDQN (Li et al., 2022) and HyperAgent (Li et al., 2024b) optimize a hypermodel that parameterizes the last layer of the Q-network for efficient exploration.

## 3 BAYESIAN ENSEMBLE FRAMEWORK

Sequential decision-making involves optimizing long-term objectives through a series of choices made under uncertainty. In Multi-Armed Bandit (MAB) problems, the goal is to maximize cumulative rewards, with actions influencing only immediate outcomes and leaving subsequent environmental states unchanged. In contrast, Reinforcement Learning (RL) tasks require accounting for dynamic state transitions while optimizing for long-term cumulative rewards. Under uncertainty, the core learning process entails inferring the underlying objective: in MAB, by estimating unknown reward distributions; in RL, by developing policies to guide long-term optimization. The exploration–exploitation dilemma remains a fundamental challenge for sequential decision-making.

Ensemble-based methods have shown their effectiveness in characterizing uncertainty. The key distinguishing feature of our proposed Bayesian Ensemble, illustrated in Figure 1, lies in its dynamic updating of the index distribution for sampling base models via Bayesian inference — unlike prior ensemble-based Thompson Sampling methods that maintain a fixed index distribution. This update directly links the index distribution to the feedback rewards through interacting with the environment, leading to a more precise approximation of the posterior distribution, and thus enhance the performance for uncertainty modeling. Bayesian Ensemble advances beyond simple ensemble aggregation techniques by introducing a learned, dynamic weighting mechanism for ensemble members. Within this framework, we elaborate on our approach as applied to sequential decision-making problems, particularly in bandit and reinforcement learning contexts.

## 4 BAYESIAN ENSEMBLE BANDIT

We first introduce a preliminary on the bandit problem setting. The problem is then addressed by our proposed Bayesian Ensemble Bandit (BEB) framework. Finally, we provide two examples on employing BEB to reinforce existing ensemble-based TS methods.

### 4.1 PRELIMINARY

We consider the multi-armed contextual bandit problem. Let $\mathcal{X}$ denote the action space, and $\mathcal{R}$ denote the reward space. At every iteration $t$, a set of actions $\mathcal{X}^{(t)} \subset \mathcal{X}$ are provided to the agent. Then, the agent selects an action $x^{(t)} \in \mathcal{X}^{(t)}$ and derives the reward $r^{(t)} \in \mathcal{R}$. Finally, $(x^{(t)}, r^{(t)})$ is utilized to update the agent policy. The objective is to maximize the cumulative rewards $\sum_t r^{(t)}$.

### 4.2 BEB FRAMEWORK

We focus on the finite reward space with $N$ distinct rewards, denoted by $\mathcal{R} = \{R_i\}_{i=1}^N$. [*] We utilize an ensemble of neural networks

$$f(x \in \cdot; z \in \cdot, \theta \in \cdot) : \mathcal{X} \times \mathcal{Z} \times \Theta \to \Delta^N \tag{1}$$

to model the probability distribution of the reward, where $x \in \mathcal{X}$ is the context vector of an action, $z \in \mathcal{Z}$ is the index of a base model, $\theta \in \Theta$ are the learnable parameters of the neural network ensemble, and $\Delta^N$ is the regular $N-1$ dimensional simplex. Each of the coordinates of $\Delta^N$ is the estimated probability corresponding to a possible reward value, i.e.,

$$\hat{\Pr}\{r = R_i\} = f(x; z, \theta)_i, \forall i \in [N]. \tag{2}$$

At each iteration $t$, the agent maintains a probability distribution $p^{(t)}$ for the index $z \in \mathcal{Z}$ and the parameters $\theta^{(t)}$. The index $z^{(t)}$ is sampled from $p^{(t)}$. Then, the agent selects the action $x^{(t)} \in \mathcal{X}^{(t)}$ that maximizes the expected reward estimated by the ensemble:

$$x^{(t)} \leftarrow \arg \max_{x \in \mathcal{X}^{(t)}} \sum_{i=1}^N R_i \cdot f(x; z^{(t)}, \theta^{(t)})_i. \tag{3}$$

and derives the reward $r^{(t)}$. Finally, the instance $(x^{(t)}, r^{(t)})$ is utilized to update the agent policy, i.e., $\theta^{(t+1)}$ and $p^{(t+1)}$. The overall algorithm Bayesian Ensemble Bandit (BEB) is shown in Algorithm 1.

---

**Algorithm 1:** Bayesian Ensemble Bandit (BEB)

---

**Input:** Initialization of learnable parameters $\theta^{(0)}$ of neural network ensemble, index distribution $p^{(0)}$.

**for** $t = 0$ **to** $T - 1$ **do**
    Receive $\mathcal{X}^{(t)}$;
    Sample $\boldsymbol{z}^{(t)} \sim p^{(t)}$ ;
    $\boldsymbol{x}^{(t)} \leftarrow \arg \max_{\boldsymbol{x} \in \mathcal{X}^{(t)}} \sum_{i=1}^N R_i \cdot f(\boldsymbol{x}; \boldsymbol{z}^{(t)}, \boldsymbol{\theta}^{(t)})_i$;
    Apply $\boldsymbol{x}^{(t)}$ and observe $r^{(t)}$;
    Update $\boldsymbol{\theta}^{(t+1)}$ and $p^{(t+1)}$;

---

While maximizing cumulative rewards is the ultimate goal in multi-armed contextual bandit problems, we often resort to minimizing a surrogate loss function when *learning model parameters*, like $\boldsymbol{\theta}$ in this case. This trade-off is primarily made due to computational efficiency, and aligns with existing ensemble-based methods. Formally, let $\ell(\cdot, \cdot) : \mathcal{R} \times \Delta^N \to \mathbb{R}$ represent the loss function,

---

[*] In fact, it is easy to see that the actual reward space could be $\mathbb{R}$; only the space over which the predicted distribution is required to be finite.

which is typically task-driven, and the binary cross-entropy loss is commonly used. Given training data $\mathcal{D} \subseteq \mathcal{X} \times \mathcal{R}$, we leverage the Empirical Risk Minimization (ERM) principle to learn $\boldsymbol{\theta}$. This involves minimizing the expected value of the empirical risk over a probability distribution $p$ governing the model index $\boldsymbol{z}$. Therefore, we estimate $\boldsymbol{\theta}$ by solving the following optimization problem:

$$\min_{\boldsymbol{\theta} \in \Theta} \sum_{(\boldsymbol{x}, r) \in \mathcal{D}} \mathbb{E}_{\boldsymbol{z} \sim p} \left[ \ell(r, f(\boldsymbol{x}; \boldsymbol{z}, \boldsymbol{\theta})) \right]. \tag{4}$$

Learning the index distribution marks a critical departure from existing ensemble-based Thompson Sampling methods. We exploit the observation that the number of parameters in $p$ (index distribution) is typically much smaller compared to $\boldsymbol{\theta}$ (ensemble parameters). This allows us to directly optimize $p$ for long-term cumulative reward, eliminating the need for surrogate losses. *Essentially, we bridge the gap between the index and reward distribution without relying on surrogate loss functions.*

While specifics depend on the chosen $p$ structure, at each iteration, probabilistic techniques like exact or approximate Bayesian inference can be employed for updating $p$. Careful design of $p$ based on the specific case is crucial for effective application.

### 4.3 Unifying Bandit Methods

Many existing ensemble-based Thompson Sampling methods can be captured as special cases of BEB. The crucial difference lies in their treatment of the index distribution $p$. Prior methods often fix $p$ throughout the decision-making process. BEB offers a powerful way to "supercharge" these methods by allowing $p$ to be dynamic and adaptive through the learning process.

Furthermore, we can categorize ensemble-based methods based on how they define the index continuity. To illustrate this, we will showcase two representative examples: a discrete and a continuous ensemble-based Thompson Sampling method. Their enhancements when combined with BEB will be further evaluated in experiments.

`ensemble+` (Osband et al., 2018) is a variation of deep ensembles (Lakshminarayanan et al., 2017) that incorporates a randomized prior function into each member. These priors, implemented as randomly initialized neural networks with identical architecture, remain fixed throughout training and inference. They have proven effective for generating uncertainty estimates. Originally, the index distribution $p$ in `ensemble+` was a discrete uniform distribution: $z \sim \mathrm{DiscreteU}([K])$, where $K$ is the number of ensemble members. Under the BEB framework, we enhance `ensemble+` by transforming $z$ to $z = \arg\max_{i \in [K]} w_i$, where the weights $(w_i)_{i=1}^{K}$ represent individual ensemble members and are drawn from Beta distributions: $w_i \sim \mathrm{Beta}(\alpha_i, \beta_i)$. This choice is motivated by the fact that the Beta distribution is the conjugate prior to the Bernoulli likelihood of the rewards, which enables efficient and exact Bayesian updates. The update rule follows Thompson sampling (Russo et al., 2018). Through Bayes' theorem, if member $i$ is selected at step $t$ and receives reward $r^{(t)} \in \{0, 1\}$, the index distribution $p$ can be updated with exact Bayesian inference: $(\alpha_i, \beta_i) \leftarrow (\alpha_i, \beta_i) + \left(r^{(t)}, 1 - r^{(t)}\right)$.

`hypermodel` (Dwaracherla et al., 2020) is a variation of deep ensembles (Lakshminarayanan et al., 2017) that treats ensemble members as specific parameter values of a base model class. It employs hypernetworks (Ha et al., 2017) to map an index to a specific ensemble member. Originally, `hypermodel` uses the multivariate standard Gaussian distribution ($z_i \sim \mathcal{N}(0, 1)$ for each $i$ in the index vector) as its index distribution. As the `hypermodel` framework maps a continuous, multi-dimensional index vector to model parameters, it is natural to model the index distribution with a multivariate Gaussian. This allows us to learn its mean and variance parameters via approximate inference, extending the original method's use of a fixed standard Gaussian. Therefore, under the BEB framework, we instead employ Gaussian distributions with individual means $\mu_i$ and variances $\sigma_i^2$ for each index component. This enables us to learn and adapt these parameters separately throughout the process. Similar to other BEB implementations, the update rule follows Thompson sampling (Russo et al., 2018). However, due to the complexity of the model, exact Bayesian inference becomes impractical. To address this, `hypermodel` utilizes variational inference (Blundell

et al., 2015) to approximate the posterior distribution by updating the means and variances of the individual Gaussian components.

## 5 BAYESIAN ENSEMBLE REINFORCEMENT LEARNING

Besides bandit problems, Reinforcement Learning represents a classic sequential decision making scenario. Prior research has investigated Ensemble-DQN (Anschel et al., 2017) and Random Ensemble-DQN (Agarwal et al., 2020), demonstrating their capability to effectively reduce the variance of Q-value estimates compared to standard DQN algorithms, which significantly stabilizes the learning process. However, we posit that the conventional practice of applying random or uniform weighting schemes across different evaluators fails to fully exploit their complementary capabilities and optimize collective performance. Similar to the idea of Self Imitation Learning (Oh et al., 2018), we propose that making better use of superior evaluators can help the agent obtain more cumulative rewards during the training process, thereby facilitating the agent to explore more effectively.

We consider the reinforcement learning (RL) framework formalized as a Markov Decision Process (MDP), defined by the tuple $\mathcal{M} = (\mathcal{S}, \mathcal{A}, P, R, \gamma)$. $\mathcal{S} = \{s_0, s_1, \ldots, s_n\}$ is the state space and $\mathcal{A} = \{a_0, a_1, \ldots, a_m\}$ is the action space, denoting the set of states and actions available to the agent. $P(s'|s, a) : \mathcal{S} \times \mathcal{A} \times \mathcal{S} \rightarrow [0, 1]$ is the transition probability function. After transitioning from $s$ to $s'$ via action $a$, the agent receives a reward $r$. $\gamma \in [0, 1)$ is the discount factor that balances the importance of immediate and future rewards. At each time step $t$, the agent observes the current state $s_t \in \mathcal{S}$, selects an action $a_t \in \mathcal{A}$ according to a policy $\pi(a|s)$, which represents the probability of taking action $a$ in state $s$. The environment then transitions to a new state $s_{t+1} \sim P(\cdot|s_t, a_t)$ and emits a reward $r_t$. The agent's objective is to learn a policy $\pi$ that maximizes the expected cumulative discounted reward: $\mathbb{E}_\pi [\sum_{t=0}^{\infty} \gamma^t r_t \,|\, s_0]$. To evaluate and improve policies, the action-value function $Q^\pi(s, a) = \mathbb{E}_\pi [\sum_{t=0}^{\infty} \gamma^t r_t \,|\, s_0 = s, a_0 = a]$. These functions satisfy the Bellman equations. In particular, the optimal action-value function $Q^*(s, a)$ satisfies the Bellman optimality equation: $Q^*(s, a) = \mathbb{E}_{s' \sim P(\cdot|s,a)} [r + \gamma \max_{a' \in \mathcal{A}} Q^*(s', a')]$. The optimal policy $\pi^*$ can be derived by acting greedily with respect to $Q^*$.

Inspired by the accelerated policy learning through multi-perspective value estimations in partially observable Markov decision processes (Li et al., 2024a), we propose Bayesian Ensemble Deep Q-Network (BE-DQN) by integrating Bayesian principles into reinforcement learning training. BE-DQN maintains $K$ independent Q-networks $\{Q(s, a; \theta_i^k)\}_{k=1}^K$ in iteration $i$ , each Q-network is parameterized by weights $\theta_i^k$. These networks are trained jointly using the standard DQN objective, but with a shared experience replay buffer.

Following the ensemble approach, we maintain a Beta distribution $w_k \sim \text{Beta}(\alpha_k, \beta_k)$ for each $k$ Q-network. At each iteration, BE-DQN samples probability $\{w_1, w_2, \cdots, w_K\}$ according to the Beta distribution of each Q-network, and selects the $j$-th Q-network with the maximum probability for action selection. At the end of each iteration, the agent receives a reward $r^{(t)} \in \{0, 1\}$. BE-DQN subsequently updates the $j$-th Beta distribution parameter through $(\alpha_j, \beta_j) \leftarrow (\alpha_j, \beta_j) + (r^{(t)}, 1 - r^{(t)})$, where the update mechanism explicitly links reward feedback with distribution adaptation.

In contrast to Ensemble DQN which estimates state action values $Q(s, a)$ through averaging outputs from multiple Q networks, BE-DQN employs a single Q-network sampling mechanism for action selection while utilizing weighted averaging across all Q-networks for objective function computation. This update mechanism effectively mitigates the overestimation bias commonly induced by individual Q-functions (Van Hasselt et al., 2016; Anschel et al., 2017; Agarwal et al., 2020), while maximizing exploitation and exploratory behavior (Ecoffet et al., 2021). Define the weight of the $k$-th model as $p_k = \frac{w_k}{\sum_{k=1}^K w_k}$, the sum weight of all model is $\sum_{k=1}^K p_k = 1$. The BE-DQN target Q network is defined as:

$$y_{s,a}^i = \mathbb{E}_\mathcal{B}[r + \gamma \max_{a'} \sum_{k=1}^K p_k Q(s', a', \theta_{i-1}^k)|s, a]. \tag{5}$$

and the Target Approximation Error (TAE) is:

$$Z_{s,a}^{i,k} = Q(s, a; \theta_i^k) - y_{s,a}^i. \tag{6}$$

---

**Algorithm 2:** Bayesian Ensemble DQN (BE-DQN)

Initialize $K$ Q-networks $Q(s, a; \theta^k)$ with random weights $\theta^k$ for $k \in \{1, ..., K\}$
Initialize Experience Replay (ER) buffer $\mathcal{B}$
Initialize exploration procedure Explore($\cdot$)
**for** *iteration $i = 1$ to $N$* **do**
   Sample the probability $\{w_1^{i-1}, \cdots, w_K^{i-1}\}$ from each distribution $\text{Beta}(\alpha_k, \beta_k)$;
   Update weights $\{p_1^{i-1}, \cdots, p_K^{i-1}\}$;
   Choose the index $j$ with the max probability $j = \arg\max_k p_k^{i-1}$;
   **for** *time $t = 1$ to $T$* **do**
      Choose action using $Q(s, a; \theta_i^j)$ ;
      $y_{s,a} = \mathbb{E}_{\mathcal{B}}[r + \gamma \max_{a'} \sum_{k=1}^K p_k Q(s', a', \theta_{i-1}^k)|s, a]$;
      **for** $k = 1$ to $K$ **do**
         $\theta_i^k \approx \arg\min_\theta \mathbb{E}_{\mathcal{B}}[(y_{s,a} - Q(s, a; \theta))^2]$;
      Explore($\cdot$), update $\mathcal{B}$;
   Update the $j$-th distribution $(\alpha_j, \beta_j) \leftarrow (\alpha_j, \beta_j) + (r^{(t)}, 1 - r^{(t)})$;
Output $Q_i^{\text{BE-DQN}}(s, a) = \sum_{k=1}^K p_k^i Q(s, a; \theta_i^k)$.

---

We provide the algorithmic implementation in Algorithm 2.

To demonstrate the effectiveness of Bayesian Ensemble DQN, we employ a simple $M$ states unidirectional MDP environment shown as Figure 2 (Anschel et al., 2017). We analyze TAE $Z_{s,a}^i$ is a random processes with specific statistical characteristics. The expected value of TAE $\mathbb{E}[Z_{s,a}^{i,k}]$ is 0, the variance of TAE is given by $\text{Var}[Z_{s,a}^{i,k}] = \sigma_s^2$, and the covariance between $Z_{s,a}^i$ and $Z_{s',a'}^j$ $\text{Cov}[Z_{s,a}^i, Z_{s',a'}^j] = 0 \quad \forall i \neq j, i \in [1, K], j \in [1, K]$. To focus exclusively on the TAEs, we eliminate the overestimation error by considering a fixed policy for updating the target values. This ensures that any bias introduced by the choice of policy does not interfere with the analysis of TAEs. By fixing the policy, we isolate the effects of temporal errors from other sources of uncertainty. Furthermore, we conveniently assume a zero reward ($r = 0$) everywhere in the environment. This assumption has no effect on the variance calculations because the reward term does not contribute to the variance of the TAEs. In this environment, the Q value of DQN (Anschel et al., 2017), for $i > M$ is:

$$Q_i^{\text{DQN}}(s_0, a) = Z_{s_0,a}^i + \gamma Z_{s_1,a}^{i-1} + \cdots + \gamma^{(M-1)} Z_{s_{M-1},a}^{i-(M-1)}, \tag{7}$$

and the variance is

$$\text{Var}[Q_i^{\text{DQN}}(s_0, a)] = \sum_{m=0}^{M-1} \gamma^{2m} \sigma_{s_m}^2. \tag{8}$$

The variance of Ensemble DQN (Anschel et al., 2017) is:

$$\text{Var}[Q_i^{\text{E-DQN}}(s_0, a)] = \frac{1}{K} \sum_{m=0}^{M-1} \gamma^{2m} \sigma_{s_m}^2. \tag{9}$$

For the variance of the BE-DQN, we establish the following two theorems.

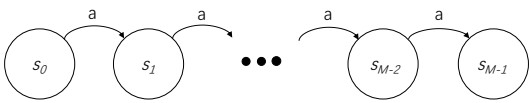

Figure 2: A $M$ states unidirectional MDP environment with a zero reward in any state.

**Theorem 1.** *The variance of BE-DQN $Var[Q_i^{BE\text{-}DQN}(s_0,a)]$ has a upper bound matching the variance of DQN, and with a lower bound matching the variance of Ensemble DQN under the same $K$, i,e., $Var[Q_i^{BE\text{-}DQN}(s_0,a)] \in [Var[Q_i^{E\text{-}DQN}(s_0,a)], Var[Q_i^{DQN}(s_0,a)]]$.*

**Theorem 2.** *The variance of each $\forall j \in [1,\ldots,K]$ BE-DQN Q-value estimator $Var[Q_i^{BE\text{-}DQN}(s_0,a,\theta_i^j)]$ has a upper bound matching the variance of DQN, and with a strict lower bound matching the variance of Ensemble DQN under the same $K$, i,e., $Var[Q_i^{BE\text{-}DQN}(s_0,a,\theta_i^j)] \in (Var[Q_i^{E\text{-}DQN}(s_0,a)], Var[Q_i^{DQN}(s_0,a)]]$.*

See Appendix A.1 for detailed proofs. Theorem 1 and 2 ensure that the variance of BE-DQN does not exceed that of standard DQN, thereby enhancing training stability and convergence — a critical property for reliable deep reinforcement learning in complex environments. Besides, compared with E-DQN, BE-DQN improves exploratory behavior by leveraging a single Q-function for action selection at each step. This design encourages temporally coherent exploration while maintaining sufficient behavioral diversity across ensemble members, effectively balancing exploration with stable exploitation.

## 6 EXPERIMENTS

Our experiments span both Neural Testbed (Osband et al., 2022) and Mushroom (Schlimmer, 1981) synthetic environment and real-world (Yahoo!R6B (Li et al., 2011)) sequential decision-making environments, echoing prior works (Osband et al., 2022; Dwaracherla et al., 2020; Osband et al., 2023; Song et al., 2021; Wu et al., 2022; Riquelme et al., 2018). In addition, we use MiniGrid (Chevalier-Boisvert et al., 2023) as the testing environment for reinforcement learning. To demonstrate the effectiveness and efficiency of our proposed BEB and BE-DQN, we compare its regret, elapsed wall time, cumulative rewards and performances against strong baselines in these diverse environments. See the Appendix for detailed experiment settings.

### 6.1 RESULTS ON SYNTHETIC ENVIRONMENTS

In synthetic sequential decision-making environments, the true reward distributions are given, which enables us to evaluate benchmark agents by calculating regrets, i.e., the cumulative reward difference from the best agent. We validate our methods on Neural Testbed (Osband et al., 2022) and Mushroom dataset (Schlimmer, 1981).

Neural Testbed (Osband et al., 2022) is an open-source library that synthesizes sequential decision-making problems by simulating an environment with data generated by neural network–based generative models. Each environment corresponds to a 2-layer fully-connected neural network with ReLU as the activation function, with independent Xavier initialization in each hidden layer, and with two nodes in the output layer, i.e., $\boldsymbol{f}^* : \mathcal{X} \to \Delta^2$; the environment serves as a ground truth for the reward distribution. At each iteration $t$, the candidate arm set $\mathcal{X}^{(t)}$ of $K$ arms are independently sampled from the standard Gaussian distribution, and the reward $r^{(t)}$ of the selected arm $\boldsymbol{x}^{(t)}$ is sampled from $\boldsymbol{f}^*(\boldsymbol{x}^{(t)})$. In our experiment, the size of the hidden layer of the generative model is set to 50, the number of actions $K = 20d$, and the time horizon $T = 20\,000$. To some extent, the dimension of arms (actions) $d$ represents the complexity of the problem. Therefore, we set the dimension of arms (actions) $d \in \{2, 10, 50\}$ (i.e., $\mathcal{X} = \mathbb{R}^d$) to show the superiority of our method over the state-of-the-arts for uncertainty modeling — `ensemble+` and `hypermodel`, in a wide range of complexity degrees.

Different from Neural Testbed, the Mushroom dataset (Schlimmer, 1981) is a real-world classification dataset that contains 8,124 instances with 22 attributes and 2 classes: poisonous and edible. We synthesize the decision-making process by letting the agent decide whether to select a given mushroom round by round following previous benchmarks (Riquelme et al., 2018; Zhang et al., 2021). The agent would receive reward 1 if an edible mushroom is selected and reward 0 otherwise. We sample 20,000 data points in the experiment.

The cumulative regrets are illustrated in Figure 3. It can be obviously observed that our models (BEB-based) outperform their corresponding baselines significantly, across all environments with different settings on the dimension throughout the whole decision-making process (at every step).

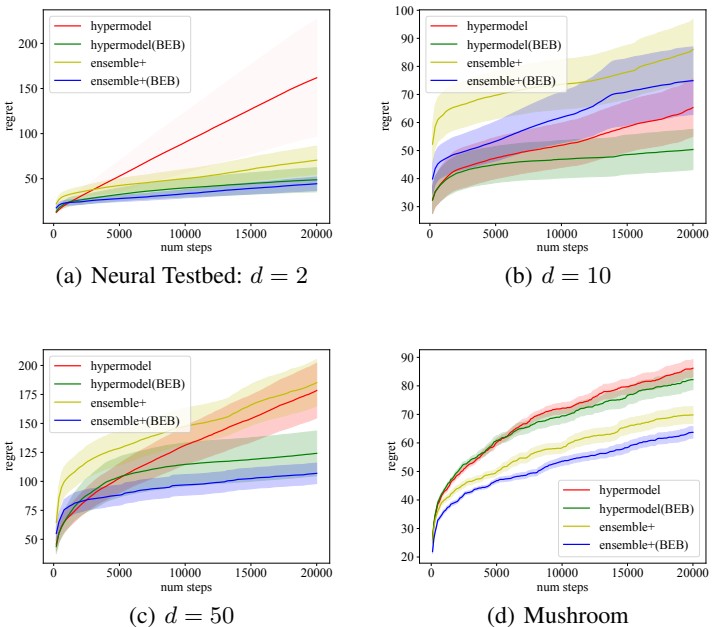

Figure 3: Regrets (the smaller the better) on Neural Testbed and Mushroom with 90% confidence interval.

Specifically, at the end of the decision-making process, for $d = 2, 10, 50$, `ensemble+(BEB)` outperforms `ensemble+` by $37.0\%$, $12.8\%$, and $42.2\%$; `hypermodel(BEB)` outperforms `hypermodel` by $69.8\%$, $22.8\%$, and $30.3\%$. The results on the Mushroom dataset are similar (also shown in Figure 3 (d)). BEB models win at every step, and at the end of the decision-making process, BEB-based `ensemble+` and `hypermodel` outperform their baselines by $8.7\%$ and $4.8\%$, respectively. These results indicate BEB is able to boost the exploration efficiency for sequential decision-making problems in these purely simulated environments.

Compared with `enemble+` and `hypermodel`, as the parameters of the index distribution are required to be updated additionally for BEB models, one may concern whether the performance gain deserves the extra time cost. Therefore, we demonstrate BEB's efficiency on Neural Testbed with different dimensions in Table 1. For `enemble+`, owing to the conjugacy property, updating the index distribution is efficient via exact Bayesian inference. Thus in all cases, the elapsed time for `enemble+(BEB)` is roughly equivalent to `enemble+`. In the case of `hypermodel`, since exact Bayesian inference is intractable, we resort to variational inference which incurs additional computational overhead – the elapsed time for `hypermodel(BEB)` increases by $20\%$. As expected, the additional overhead of BEB actually depends on the updating rule for the index distribution, which is highly related to its based Thompson sampling method.

Table 1: Elapsed wall time(s) on Neural Testbed

|  | $d = 2$ | $d = 10$ | $d = 50$ |
|---|---|---|---|
| ensemble+ | $374.33 \pm 16.47$ | $390.59 \pm 1.76$ | $1165.07 \pm 14.69$ |
| ensemble+(BEB) | $374.62 \pm 17.89$ | $389.56 \pm 1.94$ | $1162.82 \pm 38.65$ |
| hypermodel | $41.83 \pm 1.76$ | $43.28 \pm 1.68$ | $60.16 \pm 1.18$ |
| hypermodel(BEB) | $54.93 \pm 1.64$ | $55.75 \pm 1.24$ | $84.20 \pm 1.27$ |

## 6.2 RESULTS ON REAL-WORLD ENVIRONMENT

Extending our analysis, we follow prior works (Gal & Ghahramani, 2016; Song et al., 2021; Wu et al., 2022) and conduct further experiments on the Yahoo!R6B dataset (Li et al., 2011). This dataset offers 28 million user visits to Yahoo!'s Today Module over 15 days, where each visit presents four key features: (1) User Information: A 136-dimensional binary vector capturing demographics like age and gender; (2) Displayed Article: The ID of the article presented to the user; (3) Candidate Articles: A set of possible articles from which the displayed article was randomly chosen; (4) Click Label: A binary indicator of whether the user clicked the displayed article. We adopt the experimen-

tal setup of Song et al. (2021): hyperparameter tuning on day 1 entries and performance evaluation on the remaining 14 days. As ground truth rewards are unavailable, we use the cumulative number of clicks (rewards) instead of regrets.

Due to the significantly higher computational demand of `ensemble+` compared to other methods, we focus solely on the `hypermodel` in this experiment. Table 5 showcases the results on the Yahoo!R6B dataset. Notably, our method delivers a significant increase in click-through rate compared to both non-ensemble and ensemble-based methods. These compelling results demonstrate BEB's superior exploration efficiency in a real-world sequential decision-making setting, further solidifying the effectiveness of our proposed approach.

Table 2: Cumulative rewards on Yahoo!R6B

|  | Number of clicks | Relative Imp.(%) |
|---|---|---|
| Random | $25,433.5 \pm 82.0$ | $-49.45\% \downarrow$ |
| Greedy | $39,423.3 \pm 1,397.9$ | $-21.65\% \downarrow$ |
| $\epsilon-$Greedy | $43,645.0 \pm 1,407.6$ | $-13.26\% \downarrow$ |
| MC Dropout | $45,907.2 \pm 705.0$ | $-8.77\% \downarrow$ |
| Gradient TS | $47,033.7 \pm 849.0$ | $-6.53\% \downarrow$ |
| hypermodel | $49,676.8 \pm 1,355.7$ | $-1.28\% \downarrow$ |
| hypermodel(BEB) | $50,322.1 \pm 1,487.2$ | - |

## 6.3 RESULTS ON MINIGRID

To evaluate the efficacy of BE-DQN across diverse tasks, we conduct experiments in generated Box2D environments *FourRooms*, *Empty-6x6*, *LavaGapS5*, *GoToDoor-5x5* and *MultiRoom-N2-S4* using the MiniGrid framework (Chevalier-Boisvert et al., 2023). As illustrated, agents with triangular avatars operate in discrete action spaces to achieve goal-oriented objectives. These tasks require dynamic interaction with environmental objects and navigation through procedurally generated maze configurations that vary episodically.

We compare BE-DQN with four baselines: vanilla DQN (Mnih et al., 2015), Ensemble DQN (E-DQN) (Anschel et al., 2017), Random Ensemble DQN (RE-DQN) (Agarwal et al., 2020), and Unbiased Asymmetric Actor-Critic (UAAC) (Baisero & Amato, 2022). UAAC is a state-of-the-art (SOTA) method for addressing partially observable Markov decision processes. To ensure experimental rigor, we maintain consistent network architectures across all methods and report mean performance metrics with standard deviations per environment. The average score across all games is provided in Table 3 to demonstrate the overall performance for different methods. DQN exhibits limited robustness across most environments, primarily due to its reliance on single Q-value estimator. While past ensemble methods E-DQN and RE-DQN partially mitigate this issue, their performance remains unstable owing to static weighting schemes that inadequately prioritize high-quality Q-estimators. BE-DQN, in contrast to RE-DQN, dynamically updates the distribution of each Q-value estimator and further exploits the individual strengths of estimators. Although the utilization of superior estimators promotes accelerated exploration and exploitation by the agent, the variance associated with it presents challenges for policy optimization. BE - DQN effectively surmounts these limitations and stands out as a formidable competitor among the baseline methods in reinforcement learning. Also, we compare Bayesian Ensemble DQN with UAAC, BE-DQN significantly outperformed UAAC, which further validates the effectiveness of BE-DQN.

Table 3: Performance of agents on MiniGrid after $1e5$ frames training. We report the average rewards and standard error for each method. All results are averaged over 5 random seeds, with each seed tested for 100 episodes.

|  | FourRooms | Empty-6x6 | LavaGapS5-6x6 | GoToDoor-5x5 | MultiRoom-N2-S4 |
|---|---|---|---|---|---|
| vanilla DQN | $0.004 \pm 0.008$ | $0.026 \pm 0.0265$ | $0.026 \pm 0.052$ | $0.066 \pm 0.0543$ | $0.002 \pm 0.0040$ |
| Ensemble DQN | $0.012 \pm 0.0117$ | $0.162 \pm 0.1546$ | $0.178 \pm 0.1403$ | $0.12 \pm 0.0358$ | $0.042 \pm 0.0325$ |
| Random Ensemble DQN | $0.01 \pm 0.0063$ | $0.186 \pm 0.1540$ | $0.12 \pm 0.1439$ | $0.128 \pm 0.0627$ | $0.03 \pm 0.0200$ |
| Unbiased Asymmetric Actor-Critic | $0.036 \pm 0.0314$ | $0.082 \pm 0.1541$ | $0.022 \pm 0.0075$ | $0.106 \pm 0.0162$ | $0.004 \pm 0.0049$ |
| Bayesian Ensemble DQN | $0.04 \pm 0.0228$ | $0.248 \pm 0.1444$ | $0.35 \pm 0.066$ | $0.142 \pm 0.0407$ | $0.118 \pm 0.2260$ |

## 7 CONCLUSION

This paper proposes Bayesian Ensemble (BE), a unified framework that enhances ensemble-based methods for uncertainty modeling. By establishing a direct Bayesian connection between the index distribution of ensemble members and the reward distribution, BE dynamically refines both the parameters of base models and the sampling strategy governing their selection. We instantiate this framework in two critical domains, bandit learning and reinforcement learning. Extensive empirical evaluation across diverse decision-making problems demonstrates BE's effectiveness and efficiency.

## 8 REPRODUCIBILITY STATEMENT

To enhance the reproducibility of our work, we have made comprehensive efforts to document and share all relevant resources. The source code for our experiments is provided as supplementary material. Detailed experimental configurations, model hyperparameters, and procedural descriptions are thoroughly documented in the Appendix. Furthermore, complete proofs for theoretical claims and theorems presented in the paper are included in the Appendix to ensure verifiability.

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

# A APPENDIX

## A.1 THEORETICAL PROOF

In this section, we provide the detail proof of Theorem 1 and Theorem 2.

**Theorem 1.** *The variance of BE-DQN $Var[Q_i^{BE\text{-}DQN}(s_0, a)]$ has a upper bound matching the variance of DQN, and with a lower bound matching the variance of Ensemble DQN under the same $K$, i,e., $Var[Q_i^{BE\text{-}DQN}(s_0, a)] \in [Var[Q_i^{E\text{-}DQN}(s_0, a)], Var[Q_i^{DQN}(s_0, a)]]$.*

*Proof.* The Q value of BE-DQN for $i > M$ is:

$$
\begin{aligned}
Q_i^{\text{BE-DQN}}(s_0, a) &= \sum_{k=1}^{K} p_k Z_{s_0,a}^{i,k} + y_{s_0,a}^i \\
&= \sum_{k=1}^{K} p_k Z_{s_0,a}^{i,k} + \gamma \sum_{k=1}^{K} p_k Q(s_1, a; \theta_{i-1}^k) \\
&= \sum_{k=1}^{K} p_k Z_{s_0,a}^{i,k} + \gamma \sum_{k=1}^{K} p_k Z_{s_1,a}^{i-1,k} + \gamma^2 \sum_{k=1}^{K} p_k Z_{s_2,a}^{i-2,k} \\
&\quad + \cdots + \gamma^{M-1} \sum_{k=1}^{K} p_k Z_{s_{M-1},a}^{i-(M-1),k},
\end{aligned}
\tag{10}
$$

the variance is:

$$
\text{Var}[Q_i^{\text{BE-DQN}}(s_0, a)] = \sum_{k=1}^{K} p_k^2 \sum_{m=0}^{M-1} \gamma^{2m} \sigma_{s_m}^2.
\tag{11}
$$

The Upper bound of BE-DQN variance is:

$$
\begin{aligned}
\text{Var}[Q_i^{\text{BE-DQN}}(s_0, a)] &= \sum_{k=1}^{K} p_k^2 \text{Var}[Q_i^{\text{DQN}}(s_0, a)] \\
&\leq (\sum_{k=1}^{K} p_k)^2 \text{Var}[Q_i^{\text{DQN}}(s_0, a)] \\
&= \text{Var}[Q_i^{\text{DQN}}(s_0, a)].
\end{aligned}
\tag{12}
$$

Based on Cauchy–Schwarz inequality (Alzer, 1992), the lower bound of BE-DQN variance is:

$$
\begin{aligned}
\text{Var}[Q_i^{\text{BE-DQN}}(s_0, a)] &= \frac{1}{K} \sum_{k=1}^{K} 1^2 \sum_{k=1}^{K} p_k^2 \text{Var}[Q_i^{\text{DQN}}(s_0, a)] \\
&\geq \frac{1}{K} (\sum_{k=1}^{K} p_k)^2 \text{Var}[Q_i^{\text{DQN}}(s_0, a)] \\
&= \frac{1}{K} \text{Var}[Q_i^{\text{DQN}}(s_0, a)] \\
&= \text{Var}[Q_i^{\text{E-DQN}}(s_0, a)].
\end{aligned}
\tag{13}
$$

$\square$

**Theorem 2.** *The variance of each $\forall j \in [1, \ldots, K]$ BE-DQN Q-value estimator $Var[Q_i^{BE\text{-}DQN}(s_0, a, \theta_i^j)]$ has a upper bound matching the variance of DQN, and with a strict lower bound matching the variance of Ensemble DQN under the same $K$, i,e., $Var[Q_i^{BE\text{-}DQN}(s_0, a, \theta_i^j)] \in (Var[Q_i^{E\text{-}DQN}(s_0, a)], Var[Q_i^{DQN}(s_0, a)]]$.*

*Proof.* The Q value of BE-DQN $j$-th estimator for $i > M$ is:

$$
\begin{aligned}
Q_i^{\text{BE-DQN}}(s_0, a, \theta_i^j) &= Z_{s_0,a}^{i,j} + y_{s_0,a}^i \\
&= Z_{s_0,a}^{i,j} + \gamma \sum_{k=1}^{K} p_k Q(s_1, a; \theta_{i-1}^k) \\
&= Z_{s_0,a}^{i,j} + \gamma \sum_{k=1}^{K} p_k Z_{s_1,a}^{i-1,k} + \gamma^2 \sum_{k=1}^{K} p_k Z_{s_2,a}^{i-2,k} \\
&\quad + \cdots + \gamma^{M-1} \sum_{k=1}^{K} p_k Z_{s_{M-1},a}^{i-(M-1),k},
\end{aligned}
\tag{14}
$$

the variance is:

$$
\text{Var}[Q_i^{\text{BE-DQN}}(s_0, a, \theta_i^j)] = \sigma_{s_0}^2 + \sum_{k=1}^{K} p_k^2 \sum_{m=1}^{M-1} \gamma^{2m} \sigma_{s_m}^2.
\tag{15}
$$

The Upper bound of BE-DQN $j$-th estimator variance is:

$$
\begin{aligned}
\text{Var}[Q_i^{\text{BE-DQN}}(s_0, a, \theta_i^j)] &\leq \sigma_{s_0}^2 + (\sum_{k=1}^{K} p_k)^2 \sum_{m=1}^{M-1} \gamma^{2m} \sigma_{s_m}^2 \\
&= \sum_{m=0}^{M-1} \gamma^{2m} \sigma_{s_m}^2 \\
&= \text{Var}[Q_i^{\text{DQN}}(s_0, a)].
\end{aligned}
\tag{16}
$$

The lower bound of BE-DQN $j$-th estimator variance is:

$$
\begin{aligned}
&\text{Var}[Q_i^{\text{BE-DQN}}(s_0, a, \theta_i^j)] \\
&> \frac{1}{K} \sigma_{s_0}^2 + \frac{1}{K} \sum_{k=1}^{K} 1^2 \sum_{k=1}^{K} p_k^2 \sum_{m=1}^{M-1} \gamma^{2m} \sigma_{s_m}^2 \\
&= \frac{1}{K} \sigma_{s_0}^2 + \frac{1}{K} (\sum_{k=1}^{K} p_k)^2 \sum_{m=1}^{M-1} \gamma^{2m} \sigma_{s_m}^2 \\
&= \frac{1}{K} \sum_{m=0}^{M-1} \gamma^{2m} \sigma_{s_m}^2 \\
&= \frac{1}{K} \text{Var}[Q_i^{\text{DQN}}(s_0, a)] \\
&= \text{Var}[Q_i^{\text{E-DQN}}(s_0, a)].
\end{aligned}
\tag{17}
$$

$\square$

Our theoretical analysis is derived under the assumption of a zero-reward environment, where the variance of value estimates serves as a proxy for policy stability. This formulation allows us to isolate and characterize the intrinsic uncertainty arising from bootstrapping and ensemble interactions, independent of reward noise.

Under the standard assumption that $\text{Var}[Z_{s,a}^i] = \sigma_s^2$, the derivation also holds in non-zero environments. Suppose after the final state $s_{M-1}$, the agents receive a reward of 1. The Q value of BE-DQN

for $i > M$ in this environment is:

$$
\begin{aligned}
Q_i^{\text{BE-DQN}}(s_0, a) &= \sum_{k=1}^{K} p_k Z_{s_0,a}^{i,k} + y_{s_0,a}^i \\
&= \sum_{k=1}^{K} p_k Z_{s_0,a}^{i,k} + \gamma \sum_{k=1}^{K} p_k Q(s_1, a; \theta_{i-1}^k) \\
&= \sum_{k=1}^{K} p_k Z_{s_0,a}^{i,k} + \gamma \sum_{k=1}^{K} p_k Z_{s_1,a}^{i-1,k} + \gamma^2 \sum_{k=1}^{K} p_k Z_{s_2,a}^{i-2,k} \\
&\quad + \cdots + \gamma^{M-1} \sum_{k=1}^{K} p_k Z_{s_{M-1},a}^{i-(M-1),k} + \gamma^{M-1} y_{s_{M-1},a}^i,
\end{aligned}
\tag{18}
$$

Since $y_{s_{M-1},a}^i = 1$, the variance of BE-DQN remains:

$$
\text{Var}[Q_i^{\text{BE-DQN}}(s_0, a)] = \sum_{k=1}^{K} p_k^2 \sum_{m=0}^{M-1} \gamma^{2m} \sigma_{s_m}^2.
\tag{19}
$$

Theorem 1 and 2 likewise hold. Because reward terms contribute only to the mean or bias, not the variance of the Q-update. Therefore, the bounds remain valid for any bounded reward, as long as TAE variance is state-wise stationary. However, in certain environments the reward function is not defined solely as a function of the state. When the reward itself is stochastic (e.g., in multi-armed bandits where rewards follow a Beta distribution), the $1/K$ lower bound no longer holds. Nevertheless, the variance of BE-DQN remains lower than that of standard DQN. To address this limitation, which falls beyond the scope of the present work, we intend to explicitly model and bound the variance of environmental rewards in future research, integrating it into our theoretical framework.

## A.2 SYNTHETIC AND REAL-WORLD ENVIRONMENTS SETTINGS

The experiments are conducted on quad-core Intel Core i5 processor with 16GB of RAM. Each experiment is repeated 20 times with different random seeds.

We implement `ensemble+` based on deep ensembles with randomized prior functions (Osband et al., 2018). Each base model (ensemble member) is a 2-layer fully-connected neural network with ReLU as the activation function. The hidden layer size is tuned in $\{8, 16, 32, 50\}$. Each prior function (neural network) has the same configuration with the base model; its parameters are initialized by LeCun normal initialization, and kept fixed during the training process. The ensemble size is tuned in $\{10, 30, 100\}$.

We consider a linear `hypermodel` over a 2-layer fully-connected neural network with ReLU as the activation function (Dwaracherla et al., 2020). The index distribution is a standard Gaussian distribution; the index dimension is tuned in $\{6, 12, 18\}$. The hidden layer size is tuned in $\{8, 16, 32, 50\}$. The additive prior is also a linear hypermodel over an MLP with 1 hidden layer of 10 hidden units.

In the training process, the base model parameters and the index distribution (only for BEB) are updated successively every $n$ steps. A replay buffer with capacity $m$ is maintained to provide training samples. We set $n = 1$, $m = 10,000$ for Neural Testbed, $n = 1$, $m = 256$ for Mushroom and $n = 80,000$, $m = 80,000$ for Yahoo!R6B.

## A.3 MINIGRID ENVIRONMENT SETTINGS

MiniGrid (Chevalier-Boisvert et al., 2023) is a minimalist gridworld environment for reinforcement learning research, designed to study sample efficiency and generalization in partially observable Markov decision processes (POMDPs). It features a discrete action space (e.g., move, turn left or right, pick), symbolic tile-based observations, and procedurally generated levels with varying complexity (e.g., doors, keys, obstacles).

For instance, as depicted in Figure 4 (a), the agent in the *FourRooms* environment aims to reach the green goal square in the room. The agent must navigate in a maze composed of four rooms

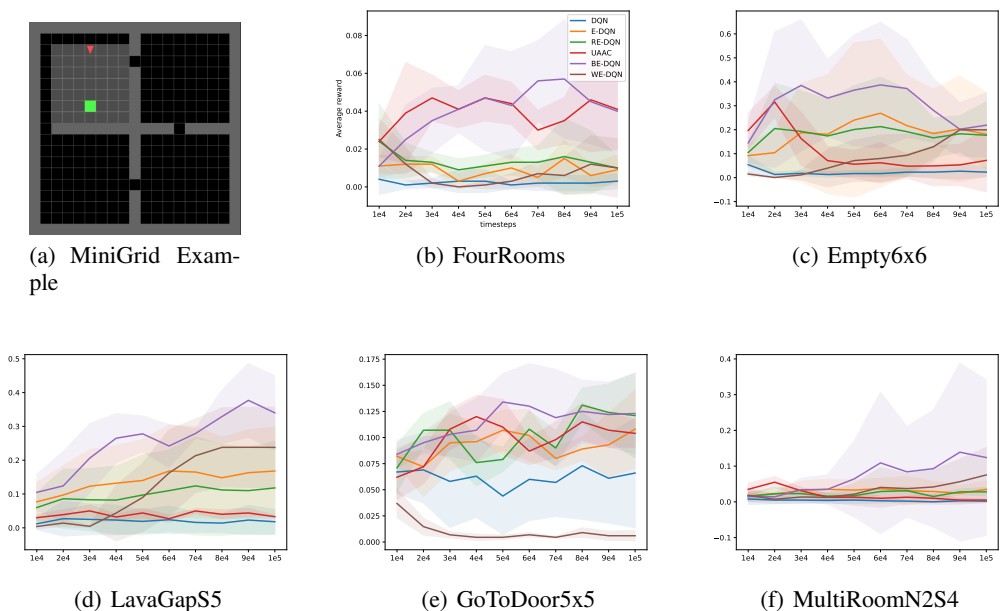

Figure 4: (a) FourRooms in MiniGrid. The agent's objective (represented by a red triangle) is to reach the target (a green box). (b-f) Performance of agents on MiniGrid after $1e5$ frames of training. All results are averaged over $5$ random seeds, with each seed tested for $100$ episodes.

Table 4: Hyperparameters used in BE-DQN.

| Hyperparameter | Value |
|---|---|
| Number of episodes | $2,000$ |
| Maximum steps per episode | $50$ |
| Ensemble size | $5$ |
| Discount factor | $0.99$ |
| Learning rate | $5 \times 10^{-4}$ |
| Batch size | $32$ |
| Replay buffer size | $5 \times 10^4$ |
| Update frequency | $1$ |
| Target network update frequency | $500$ |
| Initial $\epsilon$ | $0.1$ |
| Final $\epsilon$ | $0.02$ |
| $\epsilon$ decay factor | $0.995$ |

interconnected by $4$ gaps in the walls. To obtain a reward, the agent must reach the green goal square. Both the agent and the goal square are randomly placed in any of the four rooms. The agent receives a $7 \times 7 \times 3$ dimensional image of the environment as observation input. The three channels represent object ID, color ID, and state, respectively. The agent's position is fixed at the center of the bottom row in the observed image. The state encoding is consistent with the observation image, displaying all grids in global coordinates at all times, without rotation according to the agent's orientation. The third channel indicates that the agent is currently facing east, south, west, or north, respectively. It is worth noting that we do not impose a step penalty in the environment. Instead, we assign the reward for the entire trajectory is set to 1 if the agent achieves the goal within the allowed number of steps, and 0 otherwise.

To evaluate the efficacy of BE-DQN across diverse tasks, we conduct experiments in generated Box2D environments *FourRooms*, *Empty-6x6*, *LavaGapS5*, *GoToDoor-5x5* and *MultiRoom-N2-S4* and provide the learning curves for DQN (Mnih et al., 2015), Ensemble DQN (E-DQN) (Anschel

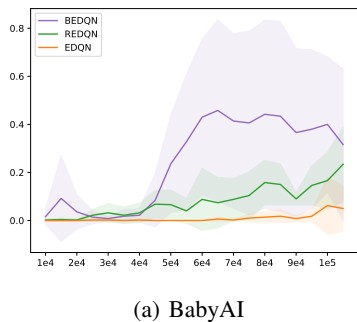
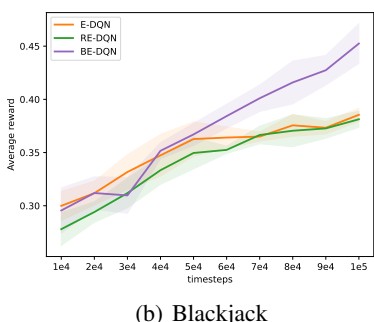

(a) BabyAI                                   (b) Blackjack

Figure 5: Performance of agents on BabyAI(*Go-To-Obj*) and Blackjack after $1e5$ frames of training. All results are averaged over $5$ random seeds, with each seed tested for $100$ episodes.

et al., 2017), Random Ensemble DQN (RE-DQN) (Agarwal et al., 2020), Unbiased Asymmetric Actor-Critic (UAAC) (Baisero & Amato, 2022) and our BE-DQN in Figure 4 (b-f). Additionally, we use a sampling approach, sampling from weights $\{p_1^{i-1}, \cdots, p_K^{i-1}\}$), named Weighted Ensemble DQN (WE-DQN). The above comparison demonstrates the effectiveness of BE-DQN.

To further validate the effectiveness of our method, we conducted experiments with E-DQN, RE-DQN, and BE-DQN in both the BabyAI and Blackjack environments, thereby confirming the efficacy of BE-DQN.

We run all experiments on a single server with AMD EPYC 7763 64-Core Processor and 1 NVIDIA P40 GPU. The hyperparameters for training BE-DQN are summarized in Table 4. The ensemble size of DQN is set to 1, while the ensemble size of DQN, E-DQN, RE-DQN, and BE-DQN is 5. The remaining parameters and network architectures are consistent. The detailed information for Q-network structure is provided below.

- **Conv2d**: (in=3, out=32, kernel=3, stride=1, pad=0)
- **ReLU**
- **Conv2d**: (in=32, out=64, kernel=3, stride=1, pad=0)
- **ReLU**
- **Conv2d**: (in=64, out=64, kernel=2, stride=1, pad=0)
- **ReLU**
- **Flatten**
- **FC**: (in=input_dim, out=512)
- **ReLU**
- **FC**: (in=input_dim, out=1)

### A.4 PERFORMANCE OF ENSEMBLE+ ON YAHOO!R6B SUBSET

Following the experimental setup outlined in Appendix A.3, we set the ensemble size to 30 and evaluated the models on a subset comprising the first 1 million events of the Yahoo!R6B dataset. The results demonstrate that `ensemble+(BEB)` outperforms the `ensemble+` baseline by a margin of 3%.

Table 5: Cumulative rewards on the Yahoo!R6B subset (first 1M events)

| Method | Number of clicks | Relative Imp.(%) |
|---|---|---|
| `ensemble+` | $2,185.4 \pm 37.65$ | $-3.1\% \downarrow$ |
| `ensemble+(BEB)` | $2,255.5 \pm 94.46$ | - |

## A.5 IMPACT OF ENSEMBLE SIZE

We investigated the impact of the number of ensemble members on `ensemble+(BEB)`. All experiments were conducted in a synthetic environment based on the Neural Testbed, with an action dimension of 50 and 10 different random seeds.

For `ensemble+(BEB)`, we evaluated ensemble sizes of $25, 50, 100$. As illustrated in Table 6, we observe a consistent improvement in performance as the number of ensemble members increases. Specifically, the relative regret reduction grows from $28.23\%$ with 25 members to $47.97\%$ with 100 members. This trend suggests that the Bayesian Ensemble framework is particularly effective at scaling with larger models.

Table 6: Relative Regret Reduction(%) of ensemble+(BEB) over ensemble+ across varying ensemble sizes

| Ensemble Size | 25 | 50 | 100 |
|---|---|---|---|
| Regret Reduction | 28.23% | 33.21% | 47.97% |

## A.6 SCALABILITY AND COMPUTATIONAL EFFICIENCY

The computational overhead of the `hypermodel(BEB)` is primarily attributed to the index updates derived from variational inference. However, a distinct advantage of our proposed framework is the inherent flexibility of this update mechanism, which allows for the modulation of update frequency to achieve an optimal balance between performance and computational efficiency.

To investigate this trade-off, we conduct experiments by adjusting the update interval. The experiments are conducted in a synthetic environment based on the Neural Testbed, with an action dimension of 50 and 10 different random seeds. Specifically, we reduce the index update frequency to one-third $(1/3)$ of the original setting. As demonstrated in Table 7, this adjustment significantly lowers the computational cost. For instance, in the 36-dimensional setting, the elapsed wall time drops from approximately 90 seconds to 77 seconds.

Crucially, the results in Table 8 indicate that this reduction in computational cost does not compromise the model's fundamental utility. While there is a trade-off, the reduced-frequency model still achieves a competitive relative regret reduction compared to the baseline hypermodel. This confirms that the BEB framework can maintain high effectiveness even under constrained computational budgets.

Table 7: Elapsed wall time(s) across different index dimensions.

| Method | Index Dim = 3 | Index Dim = 12 | Index Dim = 36 |
|---|---|---|---|
| hypermodel | $61.48 \pm 5.11$ | $61.73 \pm 2.38$ | $65.25 \pm 2.32$ |
| hypermodel (BEB) Reduced Freq. | $70.19 \pm 3.97$ | $73.59 \pm 5.09$ | $76.99 \pm 3.98$ |
| hypermodel (BEB) | $83.49 \pm 2.32$ | $86.11 \pm 3.38$ | $90.39 \pm 1.68$ |

Table 8: Relative Regret Reduction (%) compared to hypermodel across different index dimensions.

| Method | Index Dim = 3 | Index Dim = 12 | Index Dim = 36 |
|---|---|---|---|
| hypermodel (BEB) | 44.33% | 43.01% | 16.47% |
| hypermodel (BEB) Reduced Freq. | 37.98% | 32.03% | 4.37% |

