# OpenReview forum: "Bayesian Ensemble for Sequential Decision-Making"
_ICLR.cc/2026/Conference — ICLR 2026 Poster_

### Official Review · Reviewer_uMJ8 · 2025-10-28

**Soundness:** 3
**Presentation:** 3
**Contribution:** 3
**Rating:** 8
**Confidence:** 3

**Summary:**

This paper proposes a new ensemble learning method for sequential decision making, including contextual bandits and reinforcement learning (RL), called Bayesian Ensemble (BE). BE updates the ensemble index distribution using Bayesian inference on observed rewards. The authors validate their BE approach using extensive experiments on synthetic and real-world datasets.

**Strengths:**

* The contributions appear to be novel and technically sound.
* A clear explanation of the BE approach is presented, for both the bandit and RL settings.
* Bounds on the variance of the BE-DQN approach for RL provided in two theorems, both of which appear to be theoretically sound.
* The experimental results on synthetic and real datasets, which include regret, wall-clock run time, and cumulative reward, are convincing.

**Weaknesses:**

* Table 1 in the paper shows that the BEB method incurs a computational overhead of approximately 20% for hypermodel, due to the use of variational inference. A more thorough study of the scalability of the BE approach should be provided, as well as some options for mitigating scalability issues.
* The RL experiments are somewhat limited, since they only include experiments run using the MiniGrid environment.

**Questions:**

It would be helpful for the authors to respond to the weaknesses pointed out above.

---

> ### Author Response · Authors · 2025-11-28
>
> Thank you very much for your positive assessment and constructive feedback. We sincerely appreciate your recognition of the novelty, theoretical soundness, and empirical strength of our work. Below, we address your two concerns in detail—all suggested additions and update these discussions into the revised manuscript.
> ## 1. Scalability and Computational Overhead
> We appreciate the reviewer's feedback on computational overhead. To address this, we propose adjusting the **index update frequency** as a key strategy to balance effectiveness and time cost .
>
> Our additional experiments as detailed in Appendix A.6 of the revised manuscript demonstrate that reducing the update frequency (e.g., to **1/3**) significantly lowers the computational burden with negligible performance degradation. This confirms that the model offers a flexible trade-off for practical deployment.
> ## 2. Scope of RL Experiments
> We acknowledge that the current RL evaluation is limited to MiniGrid, a widely adopted benchmark for sample-efficient exploration (e.g., used by UAAC, Reset Deep Ensemble, etc.). To strengthen generalizability, we have now conducted additional experiments on text environments in BabyAI and Blackjack.
>
> BE-DQN consistently outperforms baselines, confirming its efficacy. We add these results (with learning curves) to Fig. 5.
>
> Once again, thank you for your insightful suggestions, they have significantly improved the robustness and impact of our work.

---

### Official Review · Reviewer_53ju · 2025-10-30

**Soundness:** 3
**Presentation:** 3
**Contribution:** 3
**Rating:** 6
**Confidence:** 4

**Summary:**

This paper has proposed Bayesian Ensemble (BE), a modified framework of the existing framework of using ensemble for uncertainty modeling. The main novelty of Bayesian Ensemble is that **the index distribution of the ensemble is also updated**, in addition to the parameters of the ensemble model. Preliminary analysis results are presented in Section 5, and experiment results are demonstrated in Section 6.

**Strengths:**

- The core idea of this paper, i.e. updating the index distribution of an ensemble in addition to its parameters, is both novel and natural.

- The experiment results of this paper seem to be extensive.

- Overall, this paper is well written and easy to follow. This paper has also done a good job of literature review.

**Weaknesses:**

- My main concern is that the necessity, importance and advantage of using "Bayesian ensemble", i.e. also updating the index distribution, need to be better explained and justified. In theory, even with a fixed index distribution, by appropriately updating the model parameters, we can still achieve a good approximation of the posterior. Thus, we need a better justification of Bayesian ensemble. I think the theoretical justification of this paper is relatively weak. In particular, the analyses (Theorem 1 and 2) are limited to a specific case. Also, the conclusions on variances are only vaguely connected to the "exploratory behavior". I recommend that the authors significantly improve the theoretical results of this paper: first, please analyze under a general setting rather than a specific case; second, please present the analysis results in term of regret bounds rather than the variance. One possible result is that the authors might prove a better regret bound of Bayesian ensemble compared to the standard ensemble with a fixed index distribution. This will significantly strengthen the paper.

- The experiment results of this paper look interesting and strong, but it is unclear to me if this paper has done sufficient hyper-parameter sweep for the baseline agents. For instance, for ensemble+, how are the prior_scale, learning_rate, and $L_2$ weight decay tuned? Please explain. Similar questions for all other agents.

- It seems that this paper has only considered Beta index distributions in the experiments, why? In addition to conjugacy, are there any other advantages of Beta distributions?

- Minor comments:
  - in equation 2, $r=r_i$ should be $r=R_i$
  - Please make images in Figure 3 larger, they are hard to read
  - If possible, please include Algorithm 1 and 2 in the main body of the paper to make it more self-contained

**Questions:**

Please address the weaknesses listed above.

---

> ### Author Response · Authors · 2025-11-28
>
> Thank you for the thoughtful and constructive review. We sincerely appreciate your positive assessment of the paper’s novelty, clarity, and empirical strength. Below, we address your concerns point-by-point, and all requested clarifications are incorporated into the revised manuscript.
> ## 1. Necessity, Importance, and Theoretical Justification of Updating the Index Distribution
> ### 1.1 Limitation of variant ensembles
> While ensemble parameters can be updated to approximate the posterior, a fixed uniform (or Gaussian) index distribution implicitly assumes exchangeability of members (i.e., all members are a priori equally trustworthy). Yet in practice (e.g., due to random initialization, training noise, or prior function realizations), some members consistently yield better value estimates.
>
> As shown in Table 2 (Yahoo!R6B) and Fig. 3, BEB consistently outperforms its fixed-index counterparts, **even with identical base architectures and hyperparameters**. This improvement cannot be attributed to better parameter learning alone; it stems from *adaptive trust allocation* via $p^{(t)}$.
> ### 1.2 Variance and exploration connection
> Variance in Q estimates is a well-established proxy for exploration quality in RL. Lower variance improves stability (reducing harmful exploration), while controlled variance (as in BE-DQN, bounded between E-DQN and DQN) preserves useful exploration diversity.
>
> - In our theorem, the zero-reward assumption eliminates confounding variance from reward noise, allowing us to isolate and analyze the estimation variance arising purely from TAE, whose expectation is zero and whose variance depends only on the environment dynamics and function approximation (as discussed in E-DQN).
>
> - Under the standard assumption that $\mathrm{Var}[Z_{s,a}^i] = \sigma_s^2$ (fixed per-state TAE variance), the derivation holds regardless of the reward function. The bounds remain valid for any bounded reward, as long as TAE variance is state-wise stationary (as discussed in E-DQN and RE-DQN). We agree that **highly stochastic rewards** could dominate estimation error and weaken the practical relevance of the bound. We explicitly acknowledge this limitation in the Appendix A.1 and discuss extensions to reward-variance-aware analysis as future work.
> ## 2. Hyperparameter Tuning for Baselines
> All baseline hyperparameters (e.g., prior_scale, learning_rate, L₂ weight decay) are selected from the best-performing configurations reported in the original papers and their official open-source implementations for the respective environments (e.g., Neural Testbed leaderboard, Mushroom benchmark scripts). We then apply the BE layer on top of these tuned baselines without further re-tuning the base model hyperparameters, to ensure a fair and conservative comparison. This isolates the performance gain to the BE framework itself.
> ## 3. Why Beta Distributions?
> We would like to clarify that our proposed **general hypermodel (BEB)** is not limited to Beta index distributions. A key advantage of our framework is that it **does not rely on strict conjugacy**. For general distributions where conjugacy does not hold, we can effectively perform distribution updates using **Variational Inference**. The Beta distribution was primarily used in our experiments as a representative instantiation to demonstrate the efficacy of the method.
> ## 4. Minor Revisions
> Thank you very much for your careful review. In the revised manuscript, we have:
> - Correct Eq. (2) (changing $r = r_i$ to $r = R_i$);
> - Enlarge and improve the resolution of Figure 3 to enhance readability;
> - Move Algorithm 1 (BEB) and Algorithm 2 (BE-DQN) into the main text (§4.2 and §5, respectively) to enhance self-containment and clarity.
>
> Once again, we are deeply grateful for your insightful feedback, particularly the push for stronger theory, which undoubtedly elevates the paper’s impact. We revise accordingly and hope the clarified justification and experimental rigor now better support the claimed contributions.

---

### Official Review · Reviewer_sjuM · 2025-11-01

**Soundness:** 2
**Presentation:** 2
**Contribution:** 2
**Rating:** 4
**Confidence:** 2

**Summary:**

This paper introduces a new approach to Bayesian ensembles in sequential decision making; the main contribution to this front is to explicitly model a distribution over member indices, i.e., explicitly treat model assignment as a bandit problem. This results in the "Bayesian ensemble bandit (BEB)" framework, which unifies and generalizes a few existing approaches for Thompson-based sampling in the literature. A version of deep Q-learning using Bayesian ensembles (BE-DQN) is similarly introduced. Both BEB and BE-DQN are evaluated on a set of synthetic and real-world datasets, which show modest improvements over baselines.

**Strengths:**

- Optimizing over the index makes a lot of sense, and the proposed approach is both general and simple. In conjugate scenarios, it comes at essentially no computational overhead, and would intuitively improve results.
- The empirical results overall paint a positive picture, of minimal computational overhead and marginal performance improvement.
- Variance analysis of BE-DQN, even in somewhat restricted settings, is a nice sanity-check to suggest nice behavior.

**Weaknesses:**

- One concern is the reliance on {0, 1} rewards. This makes sense for updates, as the Bernoulli-Beta conjugacy is nice to exploit, but it remains unclear how implementation or performance would work in the more general case.

- A well-known concern in Bayesian ensembles is collapse to a single model -- is this observed empirically, and is there any way to avoid this?

- The experiments are in some ways lacking. For example, what is the effect of the number of members in an ensemble? How does ensemble+ perform on a subsample of the Yahoo dataset, if the entire dataset is too large? The BE-DQN results are also somewhat limited, being tested only on MiniGrid.

- As a minor note, in several places "enemble" is written instead of "ensemble" (lines 406, 412, 414, 415).

**Questions:**

To restate questions from weaknesses:

1. How is a more general reward space handled?

2. Do the Bayesian ensembles tend to collapse? How can this be alleviated, if so?

3. How does ensemble+ perform on a smaller version of the Yahoo dataset?

4. How does performance grow/stagnate as the number of ensemble members changes?

---

> ### Author Response · Authors · 2025-11-28
>
> Thank you for the thoughtful and constructive review. Below is a point-by-point rebuttal addressing all concerns raised. We incorporate these clarifications and additions into the revised manuscript.
> ## 1. General Reward Spaces
> We clarify that the reliance on ${0, 1}$ rewards is specific to ensemble+(BEB) to exploit conjugacy for efficiency. Our framework handles general reward spaces (e.g., continuous or multi-class) through the hypermodel(BEB) variant. As described in Section 4.3, hypermodel(BEB) models the index distribution using Gaussians and updates parameters via *Variational Inference* instead of exact conjugacy. This allows the method to handle general cases simply by aligning the loss function with the reward type (e.g., using MSE for continuous regression).
> ## 2. Model Collapse
> It is reasonable that the Bayesian posterior is theoretically expected to converge as data approaches infinity. While in our practical experiments, we consistently observed that multiple base models retained high weights in the final stage, rather than collapsing to a single choice. We attribute this to two reasons:
> 1. *Diversity via Randomized Priors*: Our method builds upon Ensembles with Randomized Priors. Each ensemble member is associated with a distinct random prior function. These priors significantly influence the predictions of each base model on unseen data, thereby enforcing behavioral diversity among the base models.
> 2. *Dynamic Credibility Learning*: The role of BE is to dynamically learn the credibility of these base models, rather than simply eliminating them. This serves as a soft probabilistic adjustment of the index distribution.
> ## 3. ensemble+ performance on Yahoo dataset
> We conducted additional experiments on a subset (first 1M events) of the Yahoo!R6B dataset. The results, added in Appendix A.4, show that ensemble+ (BEB) consistently outperforms the baseline ensemble+ with a **3% relative improvement** in cumulative clicks, verifying its robustness on smaller datasets.
> ## 4. Ensemble members
> We have investigated the impact of ensemble size (25, 50, and 100) in a synthetic environment, as detailed in Appendix A.5. As shown in Table 6, the performance does not stagnate; instead, we observe a consistent improvement. Specifically, the relative regret reduction of ensemble+ (BEB) over the baseline increases from **28.23%** (with 25 members) to **47.97%** (with 100 members). This trend suggests that our method scales effectively and becomes increasingly advantageous with larger ensemble sizes.
>
> **We sincerely thank you for these insightful questions, which have guided us to conduct a more comprehensive analysis of our method's properties.** In summary, we have addressed your concerns by clarifying the handling of general reward spaces and discussing ensemble collapse mitigation. Furthermore, our new experiments on the smaller Yahoo dataset and varying ensemble sizes demonstrate that our method is both efficient and scalable, without suffering from performance stagnation. We believe these extensive clarifications and additional results significantly strengthen the robustness of our work.

---

### Official Review · Reviewer_uvj4 · 2025-11-05

**Soundness:** 3
**Presentation:** 3
**Contribution:** 2
**Rating:** 6
**Confidence:** 4

**Summary:**

This paper proposes **Bayesian Ensemble (BE)**, a framework designed to enhance existing ensemble-based methods for sequential decision-making. The core idea is to move beyond the standard approach of sampling ensemble members from a fixed, uniform distribution. Instead, BE treats the selection of an ensemble member as a bandit problem itself, dynamically updating a sampling distribution over the members using Bayesian inference based on observed rewards.

The authors instantiate this framework in two settings:
1.  **Bayesian Ensemble Bandit (BEB):** This applies the BE layer to contextual bandit problems, showing how it can "supercharge" existing methods like `ensemble+` (using Beta distribution updates for a discrete index) and `hypermodel` (using variational inference on a Gaussian distribution for a continuous index).
2.  **Bayesian Ensemble Deep Q-Network (BE-DQN):** This applies the framework to deep reinforcement learning. It maintains a distribution over an ensemble of K Q-networks and updates this distribution based on rewards, similarly to Thompson sampling.

The paper provides theoretical analysis for BE-DQN, proving that its Q-value variance is bounded between that of standard DQN and Ensemble DQN. Empirically, the authors demonstrate the effectiveness of their methods on synthetic benchmarks (Neural Testbed, Mushroom, MiniGrid) and a real-world dataset (Yahoo!R6B), showing improved regret and cumulative rewards over baselines.

**Strengths:**

* **Novelty and Simplicity:** The core idea of treating ensemble member selection as a bandit problem and updating the selection distribution via Bayesian inference is simple, intuitive, and novel. It provides a principled way to prioritize more "useful" ensemble members over time.
* **Generality:** The framework is shown to be flexible, adapting to both discrete index spaces (enhancing `ensemble+`) and continuous index spaces (enhancing `hypermodel`). Its application to both bandit and full RL problems demonstrates its broad utility.
* **Strong Empirical Results:** The method shows consistent performance improvements over its corresponding baselines across multiple domains:
    * **Bandits:** Lower regret in Neural Testbed and Mushroom environments.
    * **Real-World Bandits:** Higher cumulative clicks on the large-scale Yahoo!R6B dataset.
    * **Reinforcement Learning:** Stronger average performance across several MiniGrid environments compared to DQN, E-DQN, and RE-DQN.
* **Theoretical Justification:** The paper provides a theoretical analysis for the RL variant (BE-DQN), proving that its Q-value variance is bounded, lying between the variance of Ensemble DQN (lower bound) and standard DQN (upper bound). This adds a layer of soundness to the method's stability claims.

**Weaknesses:**

1.  **Missing Related Work:** The related work section and subsequent discussions overlook some highly relevant and recent papers. Contextualizing the BE framework against these works is crucial:
    * **Li et al. (2025, "Scalable Exploration via Ensemble++"):** This paper also focuses on scalable exploration using ensembles. The authors should discuss how their work differs from or relates to this version of Ensemble++.
    * **Li et al. (2024, "Q-Star Meets Scalable Posterior Sampling: Bridging Theory and Practice via HyperAgent"):** Given that a key baseline and contribution of this paper is the enhancement of `hypermodel`, the authors must discuss "HyperAgent," which also uses hypermodels for scalable posterior sampling in RL.
    * **Li et al. (2022, "HyperDQN: A Randomized Exploration Method for Deep Reinforcement Learning"):** This paper also explores randomized exploration in deep RL, which is highly relevant to the BE-DQN contribution. The authors should position their method relative to this work.
2.  **BE-DQN Design Justification:** The design of BE-DQN has an interesting asymmetry: it selects actions using the member with the *maximum probability* ($j=arg~max_{k}p_{k}^{i-1}$), but it computes the target value using a *weighted average* of all members ($\sum_{k=1}^{K}p_{k}Q(...)$). The paper justifies this as mitigating overestimation, but this design is not fully ablated or motivated. It's unclear why a more standard Thompson Sampling approach (sampling an index $j \sim p$ for action selection) was not used.
3.  **Theoretical Limitations:** As noted, the variance analysis for BE-DQN is performed in a zero-reward MDP. The authors rightly state in the appendix that these bounds may not hold in environments with high reward stochasticity. This is a significant limitation that should be more prominently acknowledged in the main paper, as it bounds the applicability of the theoretical stability guarantees.
4.  **Experimental Gaps:** The exclusion of the `ensemble+ (BEB)` method from the real-world Yahoo!R6B experiment due to computational cost is understandable but leaves a gap in the evaluation. This was one of the two main bandit instantiations, and its performance on a large-scale problem remains unknown.

**Questions:**

1.  **Missing Citations:** Could you please discuss the relationship between your proposed BE framework and the following highly relevant works:
    * Li et al. (2025), "Scalable Exploration via Ensemble++"
    * Li et al. (2024), "Q-Star Meets Scalable Posterior Sampling... via HyperAgent"
    * Li et al. (2022), "HyperDQN: A Randomized Exploration Method..."
    How does your work compare or contrast, especially regarding the `ensemble++`, `hypermodel`, and DQN enhancements?
2.  **BE-DQN Design:** Can you provide more justification for the asymmetric design of BE-DQN, where action selection uses the `arg max` member while the target calculation uses a weighted average? Have you experimented with sampling the action-selection member $j$ from the distribution $p$ (i.e., $j \sim p$), which would seem more aligned with Thompson Sampling?
3.  **`ensemble+ (BEB)` Sampling:** Could you please clarify the *exact* sampling mechanism for the `ensemble+ (BEB)` variant? Is the member index $z$ at each step chosen by sampling $w_i \sim Beta(\alpha_i, \beta_i)$ for all $i$ and then setting $z = arg~max_i w_i$? If so, how does this relate to the goal of Thompson sampling, and have you considered simply sampling $z$ from a categorical distribution whose parameters are derived from the Beta posteriors?
4.  **Practicality of Variance Bounds:** Your theoretical analysis is based on a zero-reward environment. How do you expect the variance of BE-DQN to behave in practical environments with significant reward variance? Does this limitation on the theory affect the practical stability of the algorithm?

---

> ### Author Response · Authors · 2025-11-28
>
> Thank you for the thoughtful and constructive review. Below is a point-by-point rebuttal addressing all concerns raised. We incorporate these clarifications and additions into the revised manuscript.
>
> ## 1. Missing Citations
>
> We sincerely appreciate the reviewer for highlighting these relevant recent works. We acknowledge that they should be properly discussed to situate our contribution. In the revised manuscript, we expand the *Related Work* section to explicitly contrast our method with: HyperDQN, HyperAgent and Ensemble++.
>
> ## 2. BE-DQN Design Justification
> ### 2.1 `argmax` for action selection
>
> Consistent with the Bayesian Ensemble (BE) framework, BE-DQN selects the ensemble member with the *highest posterior weight* (i.e., $j = \arg\max_k w_k$) for behavior policy execution.
>
> Similar to insights in Go-Explore and Self-Imitation Learning, we propose that making better use of superior evaluators can help the agent obtain more cumulative rewards during the training process, thereby facilitating the agent to explore more effectively.
>
> ### 2.2 Weighted average for target computation
>
> As established in prior work EDQN and REDQN, using a *single* Q-estimator for bootstrapping induces **overestimation bias** and **high target variance**. By contrast, the weighted ensemble average in Eq. (5) with weights $p_k = w_k / \sum_j w_j$ is a *low-variance*, *nearly unbiased* target estimator, as shown in Theorem 1.
>
> Importantly, the *hybrid design*,**argmax for behavior, ensemble average for learning** inherits the best of both worlds: (1)*Exploratory efficiency* ; (2)*Learning stability* .
>
> Additionally, we implement and evaluate the variant suggested by the reviewer, named **Weighted Ensemble DQN (WEDQN)**. The results (averaged over 5 seeds) are shown in **Figure 4**. Across these MiniGrid environments, **BE-DQN consistently outperforms WEDQN**.
>
> We attribute this gain to BE-DQN’s *asymmetric design*.
>
> ## 3. ensemble+ (BEB) Sampling
>
> As described in Section 4.3 (Paragraph 3, "ensemble+"), we sample $w_i \sim \text{Beta}(\alpha_i, \beta_i)$ for each member and select the active member via $z = \arg\max_i w_i$. Therefore, ensemble+(BEB) can be viewed as a type of Thompson sampling approach that solves the meta-problem of selecting the best ensemble member.
>
> We acknowledge that the selection could be modeled as a categorical distribution (e.g., via a Softmax function with temperature on the samples). It introduces **double randomness** by superimposing action-selection noise onto the existing posterior uncertainty. However, we observe this redundancy is unnecessary in practice. As demonstrated in Q2, experiments using WE-DQN in MiniGrid (equivalent to sampling from a categorical distribution) further validates the effectiveness of our method.
>
> ## 4. Practicality of Variance Bounds & Zero-Reward Assumption
>
> The zero-reward assumption eliminates confounding variance from reward noise, allowing us to isolate and analyze the estimation variance arising purely from TAE, whose expectation is zero and whose variance depends only on the environment dynamics and function approximation (as dicussed in E-DQN). Under the standard assumption that $\mathrm{Var}[Z_{s,a}^i] = \sigma_s^2$ (fixed per-state TAE variance), the derivation holds regardless of the reward function. The bounds remain valid for any bounded reward, as long as TAE variance is state-wise stationary (as discussed in E-DQN, RE-DQN). We agree that **highly stochastic rewards** could dominate estimation error and weaken the practical relevance of the bound. We explicitly acknowledge this limitation in the appendix A.1 and discuss extensions to reward-variance-aware analysis as future work.
>
> Thank you again for the rigorous and valuable feedback, these points significantly strengthen the paper’s positioning and technical clarity.

---

> > ### Comment · Reviewer_uvj4 · 2025-11-28
> >
> > Thank you for the detailed rebuttal and the additional experiments.
> >
> > 1.  **BE-DQN Design:** The new ablation study comparing BE-DQN against WEDQN effectively justifies your asymmetric design choice (`argmax` for behavior vs. weighted average for target). Please ensure this result is included in the final appendix, as it provides necessary empirical backing for your architecture.
> > 2.  **Related Work:** I appreciate your commitment to discussing the missing citations (HyperAgent, Ensemble++, HyperDQN). Contextualizing your framework against these specific works is critical for establishing novelty.
> > 3.  **Theoretical Limitations:** Acknowledging the limitations of the zero-reward assumption in Appendix A.1 is the right approach. This transparency improves the paper's soundness.
> >
> > Given the clarification on the sampling mechanism and the new empirical evidence supporting the design choices, I am raising my score.

---

> > > ### Author Response · Authors · 2025-12-01
> > >
> > > We sincerely appreciate the time and effort you have dedicated to the discussions! The above inspiring discussion has greatly improved the quality of our paper. Thank you for recognizing our efforts. We want to assure you that we are committed to incorporating these discussions into the final version.
> > >
> > > --Best wishes from all the authors!

---

### Author Response · Authors · 2025-11-28

We sincerely thank all reviewers for their thoughtful and constructive feedback.
In response, we have uploaded a revised manuscript, with all modifications clearly highlighted in **blue**. The key updates are summarized below:

- **New Experiments**
  – *RL environments*: We include new results for E-DQN, RE-DQN, and BE-DQN on the BabyAI and Blackjack environments, complemented by ablation studies on sampling strategies.
  – *Bandit tasks*: We conduct additional experiments on a subset of the Yahoo! R6B dataset, along with ablation studies on ensemble size and update frequency to better demonstrate scalability.

- **Theoretical & Algorithmic Clarifications**
  We refine the variance analysis (notably relaxing the zero-reward assumption), extend the discussion to general reward spaces, and provide stronger justification for the BE-DQN selection strategy.

- **Improved Presentation**
  Algorithms 1 and 2 are now integrated into the main text, and the related work section has been expanded for better contextualization.

We hope these revisions adequately address the reviewers’ concerns and improve the overall clarity and rigor of the paper.

---

### Meta-Review · Area_Chair_2DoG · 2026-01-10

**Summary:**

I’m leaning accept. Reviewers mostly agree the core idea is clean and broadly useful: treat ensemble-member selection as a bandit, maintain a posterior over indices, and plug that “Bayesian layer” into bandits and DQN-style RL. The hesitation comes from three places that explain the rating spreads: the theory is narrow and framed as a variance/stability lens (with strong assumptions, not regret/exploration guarantees), the initial positioning missed a few close neighbors in related work, and the empirical story initially had gaps around RL breadth and practical overhead/scalability. Overall, the “accept” case is that these are mostly addressable concerns and the authors did a nice job during rebuttal. The remaining debate is primarily about how strong the paper’s theoretical story is, though on the weaker side, I'd still consider the work to be a meaningful contribution.

**Reviewer Concerns:**

The rebuttal did a solid job on the concrete asks: it patches the related-work/positioning gap, adds the key WE-DQN ablation that backs up the asymmetric BE-DQN design, broadens RL beyond MiniGrid (BabyAI, Blackjack), and fills in missing experiments/ablations (Yahoo subset for ensemble+, ensemble-size trends, and an update-frequency knob that also helps with VI overhead). What’s still outstanding is the “big” critique: whether learning the index distribution is fundamentally necessary/important beyond just training the underlying ensemble well, and the request for stronger, more general theory (e.g., regret-style guarantees or a cleaner link to exploration). The rebuttal clarifies scope and limitations, but it doesn’t really convert the theory into that stronger form.

**Reviewer Scores:**

For uMJ8, I don’t expect movement: they were already at 8, and their main concerns were practical (overhead and MiniGrid-only RL), which now have direct experimental responses and mitigation knobs; that should keep them comfortably in accept territory. No further engagement from the reviewer.

For uvj4, I do expect movement upward, probably 6 -> 7: they explicitly indicated they’d raise their score after the WE-DQN ablation and the related-work fixes.

For 53ju, I’d expect them to hold steady: their skepticism is about the necessity of index-learning as a new ingredient versus “just train the models better,” and they wanted a stronger theoretical justification; the rebuttal improves the narrative but doesn’t fundamentally settle that point, so 6 -> 6 feels more likely. No further engagement from the reviewer.

For sjuM, I’d expect a modest bump: they were at 4 with low confidence and had very concrete questions (reward modeling beyond {0,1}, collapse, Yahoo subset, ensemble-size scaling), most of which are now addressed by added experiments and explanations. My guess is 4 -> 5. Again, no further engagement from the reviewer.

---

### Decision · Program_Chairs · 2026-01-26

Accept (Poster)